# Identification of a Novel Eight-Gene Risk Model for Predicting Survival in Glioblastoma: A Comprehensive Bioinformatic Analysis

**DOI:** 10.3390/cancers15153899

**Published:** 2023-07-31

**Authors:** Huy-Hoang Dang, Hoang Dang Khoa Ta, Truc Tran Thanh Nguyen, Chih-Yang Wang, Kuen-Haur Lee, Nguyen Quoc Khanh Le

**Affiliations:** 1International Ph.D. Program for Cell Therapy and Regeneration Medicine, College of Medicine, Taipei Medical University, Taipei 110, Taiwan; d151109003@tmu.edu.tw; 2Ph.D. Program for Cancer Molecular Biology and Drug Discovery, College of Medical Science and Technology, Taipei Medical University and Academia Sinica, Taipei 110, Taiwan; d621109004@tmu.edu.tw (H.D.K.T.); chihyang@tmu.edu.tw (C.-Y.W.); khlee@tmu.edu.tw (K.-H.L.); 3Graduate Institute of Cancer Biology and Drug Discovery, College of Medical Science and Technology, Taipei Medical University, Taipei 110, Taiwan; 4Taiwan International Graduate Program in Interdisciplinary Neuroscience, National Taiwan University and Academia Sinica, Taipei 115, Taiwan; m610109011@tmu.edu.tw; 5Cancer Center, Wan Fang Hospital, Taipei Medical University, Taipei 110, Taiwan; 6Professional Master Program in Artificial Intelligence in Medicine, College of Medicine, Taipei Medical University, Taipei 110, Taiwan; 7AIBioMed Research Group, Taipei Medical University, Taipei 110, Taiwan; 8Research Center for Artificial Intelligence in Medicine, Taipei Medical University, Taipei 110, Taiwan; 9Translational Imaging Research Center, Taipei Medical University Hospital, Taipei 110, Taiwan

**Keywords:** differentially expressed genes, genetic biomarkers, glioblastoma, prognosis-related genes, survival analysis, univariate cox regression

## Abstract

**Simple Summary:**

People with glioblastoma (GBM) universally have poor survival despite undergoing aggressive treatments. In this study, we aimed to determine genetic biomarkers of GBM that exhibit prognostic implications and examine their role in the tumor microenvironment. To this end, we performed differential gene expression analysis in three independent GBM datasets, followed by establishing a risk model for disease progression. Containing eight genes, this model demonstrated robustness in identifying patient subgroups with poor survival outcome in independent datasets.

**Abstract:**

Glioblastoma (GBM) is one of the most progressive and prevalent cancers of the central nervous system. Identifying genetic markers is therefore crucial to predict prognosis and enhance treatment effectiveness in GBM. To this end, we obtained gene expression data of GBM from TCGA and GEO datasets and identified differentially expressed genes (DEGs), which were overlapped and used for survival analysis with univariate Cox regression. Next, the genes’ biological significance and potential as immunotherapy candidates were examined using functional enrichment and immune infiltration analysis. Eight prognostic-related DEGs in GBM were identified, namely *CRNDE*, *NRXN3*, *POPDC3*, *PTPRN*, *PTPRN2*, *SLC46A2*, *TIMP1*, and *TNFSF9*. The derived risk model showed robustness in identifying patient subgroups with significantly poorer overall survival, as well as those with distinct GBM molecular subtypes and *MGMT* status. Furthermore, several correlations between the expression of the prognostic genes and immune infiltration cells were discovered. Overall, we propose a survival-derived risk score that can provide prognostic significance and guide therapeutic strategies for patients with GBM.

## 1. Introduction

Glioblastoma (GBM) is one of the most common central nervous system cancers, accounting for nearly half of all malignant tumors [1]. The prognosis of GBM is especially poor, with median overall survival of only 16 months. Most patients relapse even after undergoing aggressive therapies, including surgical resection, radiation, and systemic therapy [2,3]. Ongoing and past clinical trials for recurrent GBM have used a wide range of novel treatments, from cytotoxic chemotherapeutic agents to immunotherapy and gene therapy [4]. However, the substantial heterogeneity of patients in these trials could arguably preclude the generalizability of the interventions, and a better understanding of the molecular features of GBM is needed [2].

Gene expression profiling is one of the most important tools in cancer research [5]. With the identification of patterns of genes expressed at the transcriptional level, it has been leveraged to discover novel prognostic markers and treatment targets. A number of gene expression profiling-derived models has been recently developed for GBM [6,7,8,9,10]. For example, Kawaguchi et al. used random survival forests, a machine learning algorithm, to develop a 25-gene signature that was consistently correlated with overall survival in GBM [8]. Similarly, Cao et al. obtained gene expression data of GBM patients from public datasets and found four significant overlapped differentially expressed genes (DEGs), which was further validated using cell cultures and Western blots of in-house GBM specimens [6]. More recently, Wen et al. built a seven-gene, hypoxia-based prognostic signature that showed high sensitivity and specificity in predicting overall survival and chemotherapeutic response in GBM patients [9].

While these models have been successful in assessing prognosis, their clinical application faces multiple challenges. Our study aims to extend their approach by developing a gene signature with implications in the prognosis of GBM, integrating multiple databases and analyses. We further examined its biological significance and signaling pathways, with the purpose of clarifying how the model can serve as a biomarker in the prognosis and progression of GBM.

## 2. Materials and Methods

### 2.1. Data Acquisition and Preprocessing

First, we obtained gene expression from GBM specimens from three publicly available datasets, the TCGA-GBM (The Cancer Genome Atlas) and the gene expression profiles GSE4290 and GSE68848, from the Gene Expression Omnibus (GEO) database [11,12,13,14]. The GSE4290 contained 81 samples of GBM and 23 samples from epilepsy patients as non-tumor samples, detected by the Affymetrix Human Genome U133 Plus 2.0 Array based on the GPL570 platform [14]. Data of astrocytoma and oligodendroglioma samples in GSE4290 were not included in this study. The GSE68848 expression profile contained 580 samples, including 228 GBM and 28 control tissues, detected by the same platform as GSE4290 [12]. Data for other disease types (oligodendroglioma, astrocytoma, mixed, unclassified and unknown) in GSE68848 were also not included in this study. We further included gene expression RNAseq data of normal brain frontal cortex samples (BA9) from the Genotype-Tissue Expression (GTEx) database. The characteristics of the included datasets are given in Table 1. For the TCGA-GBM data, we only included the data of participants with isocitrate dehydrogenase (*IDH*)-wildtype and omitted those of *IDH*-mutant patients. This was to accommodate the 2021 WHO classification of tumors of the central nervous system, which defined GBM as strictly *IDH*-wildtype [15]. *IDH* mutation status was not available in the GSE4290 and GSE68848 datasets.

Prior to conducting further analyses, we performed data preprocessing to ensure the quality and reliability of the gene expression profiles. This involved the application of principal component analysis (PCA) to identify and remove potential outliers in the datasets. PCA is a dimensionality reduction technique commonly used in gene expression analysis to identify patterns and variability within high-dimensional datasets [16]. Outliers could arise due to various factors such as technical artifacts or experimental variability. These outliers can have a significant impact on downstream analyses and may lead to erroneous interpretations. By performing PCA, we aimed to identify any extreme or deviant samples that might introduce noise or bias into subsequent analyses.

### 2.2. Analysis of Differentially Expressed Genes (DEGs)

To determine the molecular alterations associated with GBM, we performed a differential gene expression analysis. Specifically, genes expressed differentially between GBM and the control samples were identified, using the R package *DESeq2* for TCGA-GBM versus GTEx, and the R package limma for two GEO datasets, GSE4290 and GSE68848 [17,18]. The limma package is a widely used tool in genomics research due to its robustness and flexibility in detecting DEGs. Genes were considered significant if they showed an absolute log fold change greater than 1, indicating a substantial change in expression levels, and an adjusted *p*-value less than 0.05, reflecting statistical significance after correcting for multiple testing. Then, we overlapped the DEGs lists from three comparisons to identify common genes for further exploration.

### 2.3. Functional Enrichment Analysis of the DEGs

Functional annotation and enrichment analysis are commonly used bioinformatic tools to help attribute biological information and significance to a group of genes. With the common DEGs determined, we performed Gene Ontology (GO) and Kyoto Encyclopedia of Genes and Genomes (KEGG) pathway enrichment analyses using the R package *gProfiler* to identify the DEGs pathways [19]. GO is the standardized vocabularies of genes and their products, consisting of three independent semantics: biological processes, cellular components, and molecular functions [20]. The Kyoto Encyclopedia of Genes and Genomes (KEGG) is a knowledge hub for gene function analysis, linking genes to higher-order functional information [21]. *p* < 0.05 was considered the statistically significant threshold.

### 2.4. Risk Score Construction and Validation

We next obtained survival data of GBM patients and used univariate Cox regression analysis on the 1934 DEGs. This was performed using the TCGA-GBM dataset with *IDH*-wildtype participants, as survival data were not available in GSE4290 and GSE68848. Genes significantly correlated with overall survival with *p* < 0.001 were selected. Next, we calculated the risk score for each participant as follows:(1)∑Gene Expression ∗ Coefficient

Participants were subsequently classified into the high- and low-risk group based on their risk score being higher or lower than the median cutoff value. To validate the predictive power of this prognostic model beyond the TCGA-GBM dataset, we applied Formula (1) for GBM patients from the Chinese Glioma Genome Atlas (CGGA) database and from the GEO dataset GSE43378 [8,22]. The GSE43378 dataset contains gene expression and survival data of 32 patients with GBM. Furthermore, using the R package *timeROC*, we tested the performance of the risk model as a marker for GBM over time in the TCGA-GBM dataset [23]. In contrast to the standard receiver operating characteristic (ROC) curve analysis which regards the marker value and disease status of a person as fixed for the whole study period, time-dependent ROC allows these variables to change over time, which better reflects conditions in real life [24].

### 2.5. Characterization of the Risk Model-Based Subgroups

We next investigated the relationship between the risk score and clinical characteristics, including *MGMT* status and between tumor-intrinsic gene expression subtypes of GBM [25]. *MGMT* encodes the enzyme O^6^-alkylguanine DNA alkyltransferase that is responsible for DNA repair following alkylating agent chemotherapy. As such, methylation of the *MGMT* promoter leads to loss of expression of the *MGMT* DNA repair protein, which predicts a benefit from chemotherapy and improved survival in GBM [26]. It is considered to be a highly recommended key molecular diagnostic test in GBM. Tumor-intrinsic subtypes of GBM included the proneural, classical, mesenchymal, and neural subtypes, which were previously identified from data of 200 GBM samples [27]. This classification model has since been used extensively to find distinct responses to treatment options and revised to ensure that all subtypes contain actual tumor cells [28].

We then predicted the upstream regulators (transcription factors, TFs) of the risk-related gene candidates using NetworkAnalyst (version 3.0, http://www.networkanalyst.ca, accessed on 24 June 2023) [29]. TF–gene interaction analysis was performed with the ChIP Enrichment Analysis (ChEA) database [30].

Next, we performed gene set variation analysis (GSVA) to compare metabolic signatures between the high and low risk groups. GSVA is a computational method that can estimate pathway and bioprocess activity scores from gene expression data [31]. Specifically, we employed the GSVA R package GSVA, implementing the single-sample gene set enrichment analysis (ssGSEA) method [31,32,33].

### 2.6. Relationship between the Prognostic Genes and Immune Infiltration in GBM

Finally, we performed immune cell infiltration analysis using the CIBERSORT and MCP-counter (Microenvironment cell populations-counter) methods. In the TCGA dataset, CIBERSORT was used to estimate the proportion of 22 immune cell types, and the correlation between the expression level of genes in the risk score and the abundance of the immune cells was calculated [34]. Similarly, MCP-counter enables the quantification of the absolute abundance of eight immune and two stromal cell populations in GBM tissues from transcriptomic data [35].

## 3. Results

### 3.1. Identification of DEGs from GBM Datasets

Our study obtained microarray data of GBM and control specimens from four datasets: TCGA-GBM, GTEx, GSE4290 and GSE68848. Data preprocessing with PCA identified six potential outliers in the GSE4290 dataset, which were removed from subsequent analyses (Appendix A). Using the cutoff criteria of absolute log fold change greater than 1 and adjusted *p* value less than 0.05, we identified 11,769 DEGs from TCGA-GBM versus GTEx, including 7121 upregulated and 4648 downregulated genes. GSE4290 had 3860 DEGs, with 2031 upregulated and 1829 downregulated ones. There were 3277 DEGs in GSE68848, including 1544 upregulated and 1733 downregulated genes. By overlapping these DEGs, we found 1934 genes that were significantly differentially expressed among the four datasets (Figure 1, Appendix A).

### 3.2. Functional Enrichment Analysis of DEGs

We performed GO enrichment analysis to explore the biological processes (BP), cellular components (CC), and molecular functions (MF) associated with the DEGs, separately for those that were consistently upregulated (939 DEGs) or downregulated across the three comparisons of DEG analysis (733 DEGs).

The ten most enriched GO terms for the upregulated DEGs are shown in Figure 2A–C. We found that they were mostly involved in the extracellular matrix (ECM), as evident from the most significant GO terms (BP: extracellular matrix and structure organization, CC: collagen-containing ECM, MF: ECM structural constituent). This is consistent with KEGG results, which showed extracellular matrix–receptor interaction as one of the most significantly enriched pathways (Figure 2D).

Downregulated DEGs, on the other hand, were mostly enriched in the regulation of synaptic structure and function, as seen from the GO terms (BP: modulation of chemical synaptic transmission, regulation of trans–synaptic signaling and synaptic plasticity; CC: synaptic membrane, glutamatergic synapse; MF: gated and ion channel activity) (Figure 3A–C). This is also in line with the KEGG results, which revealed GABAergic synapse, synaptic vesicle cycle, and calcium signaling pathway as the three most significant pathways (Figure 3D).

### 3.3. Identification of Prognosis-Related Genes and Construction of Risk Model

To investigate the prognostic value of the DEGs, we applied univariate Cox regression analysis using overall survival data from the TCGA-GBM dataset with a cutoff of a p-value less than 0.001. This results in eight genes, namely *CRNDE*, *NRXN3*, *POPDC3*, *PTPRN*, *PTPRN2*, *SLC46A2*, *TIMP1*, and *TNFSF9* (Table 2). Statistical values of all 1934 DEGs are provided in Appendix A. Next, a risk score of disease progression for each patient was calculated using the gene expression and coefficient of the eight genes as the following formula:Risk score = (0.00047 × *CRNDE*) + (0.00067 × *NRXN3*) + (0.00202 × *POPDC3*) + (0.00017 × *PTPRN*) + (0.00012 × *PTPRN2*) + (0.06594 × *SLC46A2*) + (9.03 × 10^−6^ × *TIMP1*) + (0.00361 × *TNFSF9*)

Patients were subsequently categorized into the high- and low-risk groups based on their risk score greater or less than the median value, respectively. As shown in Figure 4A, the Kaplan–Meier plot of the eight-gene risk model reveals that the high-risk group had significantly poorer overall survival compared to the low-risk group (*p* < 0.0001). To further validate the predictive value of the model, a similar analysis was performed on two independent datasets, CGGA and GSE43378, which interestingly showed consistent results with those from TCGA-GBM (*p* < 0.005 for all two datasets, Figure 4B,C). In addition, the time-dependent ROC curve of the risk model to predict 1-, 3-, and 5-year overall survival of GBM patients in TCGA-GBM is presented in Figure 4D. It can be seen from the reported area under the ROC (AUC) curve that the model performed increasingly better over the years, with 1-, 3-, and 5-year AUCs of 0.76, 0.65, and 0.7, respectively. This signifies that our eight-gene risk model could be a reliable tool to assess prognosis in GBM.

### 3.4. Characteristics of the Eight-Gene Risk Model

As our eight-gene risk model was able to prognostically classify patients with GBM, we next sought to examine its correlation with other clinical characteristics of GBM, including molecular subtypes and *MGMT* status. Figure 5A,B show that the risk score in *MGMT*-methylated GBM patients was significantly lower than that of *MGMT*-unmethylated patients in both the TCGA and CGGA datasets (*p* value 0.03 and 0.007, respectively), which is consistent with the fact that patients with *MGMT*-unmethylated tumors have poorer prognosis and are less responsive to standard therapies [36]. This suggests that while *MGMT* was not among the genes in our risk model, it could partly explain the model’s prognostic role. We further compared the risk score in different molecular subtypes of GBM, in which we excluded data for the neural subtype, as this subtype was found to have a high content of non-tumor cells [25,28]. Figure 5C shows that the risk score did not differ significantly between the proneural, mesenchymal and classical tumors, consistent with previous reports that there were no significant differences in survival outcome between these three subgroups [25].

Using NetworkAnalyst, we observed a TF–gene biomarker regulatory network including 143 interaction pairs among 7 seed genes (in red) and 86 TFs (in purple) (Appendix A). Specifically, *PTPRN2* was regulated by the most TFs (38 TFs), followed by *NRXN3* and *TNFSF9* (27 and 21 TFs, respectively).

Finally, we investigated the tumor-infiltrating immune cell profile of our risk model. Using the CIBERSORT method, we found that the expression of 4 out of the 22 immune cells was significantly different between the low- and high-risk groups, including the regulatory T cells, activated natural killer cells, activated dendritic cells, and neutrophils (Figure 6A). Remarkably, analysis of immune infiltration using the MCP-counter method showed that 9 out of 10 tissue-infiltrating immune and stromal cell populations were more abundant in the high-risk group (Figure 6B). Similarly, nearly all tumor microenvironment, tumor signature and epithelial–mesenchymal transition (EMT) signatures were significantly higher in the high-risk group than the low-risk group (Figure 6C–E). This suggests that our eight-gene prognostic signature is considerably correlated with measures of tumor immune infiltrates in GBM.

## 4. Discussion

GBM is the most prevalent and progressive brain malignancy. People with GBM reportedly have a median overall survival of only 16 months, despite undergoing an aggressive array of treatments, including surgery, radiotherapy and chemotherapy [2]. In this study, we performed differential gene expression analysis in three independent GBM datasets, followed by establishing a risk model for disease progression using clinical data obtained from the TCGA database. This risk model contained eight genes and was further validated in two other GBM datasets, which demonstrated its robustness in identifying patient subgroups with prognostic implications. Furthermore, significant differences in the risk score were also identified in patients with distinct GBM molecular subtypes and *MGMT* status.

Our study aims to determine prognostic markers of GBM, with the identification of DEGs followed by survival analysis. The concept of DEGs stems from the availability of the expression of mRNA transcripts, which allows the identification of genes that are significantly differentially expressed across two or more conditions—in this context, GBM and normal controls. It is one of the most common steps in analyzing microarray and RNA-seq data [37]. However, critics have argued that while DEGs can help to serve as clinical biomarkers or obtain mechanistic insights into diseases, they do not necessarily represent causes between gene expression and phenotypes, and rather could be consequences or simply correlations [38]. There are a few approaches to circumvent this issue. For instance, baseline characteristics of participants, such as age, sex, and treatment status, are known to be a significant source of variations in differential expression testing [38,39]. Incorporating them into the DEG analytical workflow was unfortunately not possible in our study due to the lack of clinical data in the GEO datasets. Alternatively, one could start with a hypothesis about a gene or set of genes of interest, e.g., whether they are related to a pathway known to drive cancers, as was performed in our previous publications [40,41].

Our risk model was constructed based on eight genes which have been individually revealed to play a wide range of roles in the development and prognosis of GBM. *CRNDE* (colorectal neoplasia differentially expressed), whose transcripts were categorized as long non-coding RNAs, was significantly overexpressed in glioma tissues compared to control brain tissues [42,43]. Zhao et al. further reported that knockdown of *CRNDE* improves sensitivity to temozolomide, a first-line chemotherapy treatment for GBM, by regulating autophagy [44]. *NRXN3* (neurexins 3) belongs to the Neurexins family, which are neuronal cell surface proteins, and play roles in cell adhesion and recognition [45]. It was shown to be downregulated in gliomas and inhibited the invasion and migration of tumor cells [46]. *NRXN3* expression is directly regulated by Forkhead box Q1, a member of the Fox transcription factor family that regulates the cell cycle, leading to promotion proliferation and the migration of GBM cells in vitro [45]. *PTPRN* (tyrosine phosphatase receptor type N) is highly expressed in endocrine cells and neuroendocrine neurons, including the pituitary, amygdala and hypothalamus. Wang et al. demonstrated that *PTPRN* overexpression promoted the migration and proliferation of glioma cells via activating the PI3K/AKT pathway, an intracellular signaling pathway known for regulating the cell cycle.

*TIMP1* (tissue inhibitor of matrix metalloproteinase 1) was shown to be correlated with cancer progression, specifically in GBM, and is significantly overexpressed in tumor-infiltrating lymphocytes [47]. As its name suggests, TIMP1 protein could inhibit the proteolytic activity of matrix metalloproteinases, which are endopeptidases that degrade the extracellular matrix, thus explaining its role in metastasis [48]. *TNFRSF9* (tumor necrosis factor receptor superfamily member 9, also known as *CD137*) is a costimulatory transmembrane protein expressed on leukocytes that stimulates B cell antibody secretion and T-cell proliferation [49]. Thus, its role as a possible immunotherapy option for GBM has been explored, with Blank et al. reporting a novel *TNFRSF9*-positive reactive astrocytic phenotype in human gliomas [50]. In another study, *TNFRSF9* was included in three immune-related genes’ signatures that could serve as independent prognostic factors for GBM patients [51]. The role of the remaining genes in the risk model, *POPDC3* (Popeye Domain Containing 3), *PTPRN2* (Protein Tyrosine Phosphatase Receptor Type N2), and *SLC46A2* (Solute Carrier Family 46 Member 2,) in GBM remains largely unknown, though each has been found to be potential biomarkers in other types of cancer [52,53].

The main limitation of this study is that data were largely procured from online databases. Part of the DEG analysis was performed using microarray data from GEO datasets, which are known to be inferior in the quality of transcriptome profiling compared to RNA-Seq data [54]. Nonetheless, the important steps in our analytical pipeline, specifically the DEG analysis and inquiry of prognostic implications of the model, were constructed using multiple independent datasets, hoping to offset the likelihood of false discoveries. Importantly, the preliminary nature of the findings encourages further experimental validations to confirm them. One other relevant implication is to consider the application of public databases in light of the latest WHO classification of tumors of the central nervous system in 2021. The entity that was previously *IDH*-mutant GBM is now referred to as *IDH*-mutant astrocytoma WHO grade 4, owing to the discovery of the significance of *IDH* mutations in the prognosis of diffuse gliomas [15]. Specifically, *IDH*-mutant tumors have a better prognosis than that of *IDH*-wildtype tumors, across all histologic grades [55]. Understandably, *IDH* status is not available in all GEO datasets of GBM patients, and thus the derived analysis would unavoidably include a subset of participants with its mutations. While it would take time for datasets collected after the 2021 WHO classification to include only those with *IDH*-wildtype status as GBM and be increasingly available, for the current study we strived to include data of only *IDH*-wildtype GBM patients where applicable.

## 5. Conclusions

Our study proposed a prognosis-derived risk score that can have prognostic implications for patients with GBM, which was validated in three independent datasets. The findings could possibly shed light on future treatment strategies for this progressive disease.

## Figures and Tables

**Figure 1 cancers-15-03899-f001:**
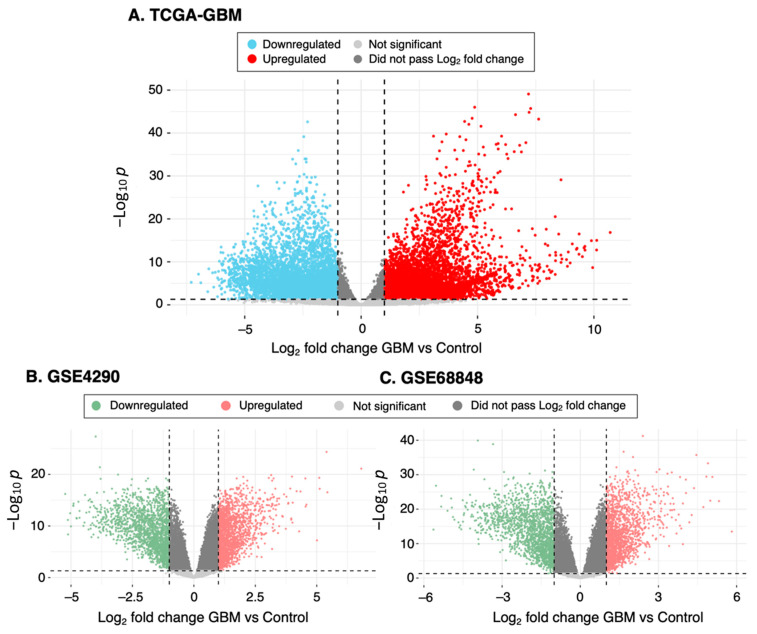
Summary of the DEGs. Volcano plot showing differential expression analysis of genes between GBM and control samples in (**A**) TCGA-GBM versus GTEx, (**B**) GSE4290 and (**C**) GSE68848. Red and blue (or green) points indicate genes with significantly increased or decreased expression, respectively, in GBM specimens compared to controls. The Log_2_-fold differences between GBM and controls are plotted on the horizontal axis, while the −Log_10_
*p*-value differences are plotted on the vertical axis. The horizontal dashed line represents the significance threshold (*p* value < 0.05 after correcting for multiple comparisons).

**Figure 2 cancers-15-03899-f002:**
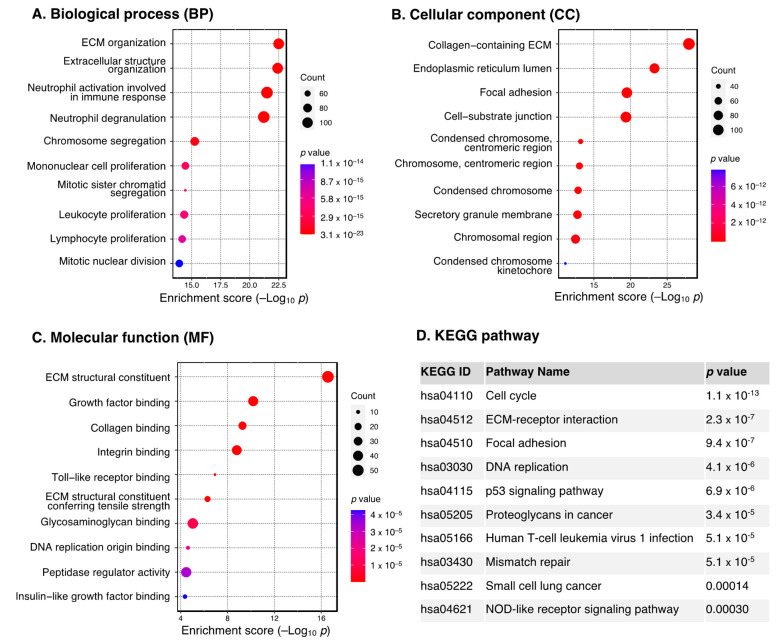
GO and KEGG pathway enrichment analysis of the upregulated DEGs. Dot plot of enriched GO terms: (**A**) biological processes, (**B**) cellular components and (**C**) molecular functions. GO processes are ordered according to the enrichment score. Dot size represents the number of genes in the significant gene list associated with the GO term. Dot color represents the adjusted *p* values. (**D**) Pathway enrichment analysis in the KEGG database. Pathways significantly enriched by DEGs are ordered by adjusted *p* values. Abbreviations: DEGs, differentially expressed genes; ECM, extracellular matrix; GO, Gene Ontology; KEGG, Kyoto Encyclopedia of Genes and Genomes.

**Figure 3 cancers-15-03899-f003:**
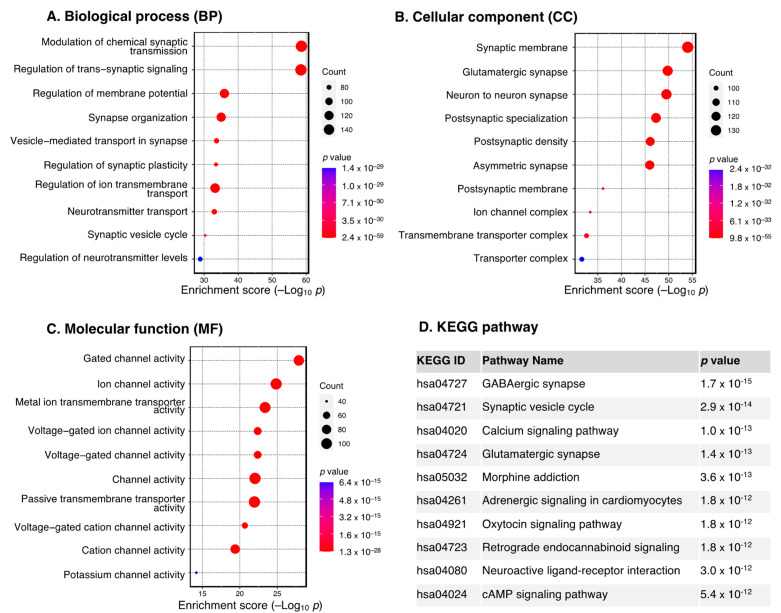
GO and KEGG pathway enrichment analysis of the downregulated DEGs. Dot plot of enriched GO terms: (**A**) biological processes, (**B**) cellular components and (**C**) molecular functions. GO processes are ordered according to the enrichment score. Dot size represents the number of genes in the significant gene list associated with the GO term. Dot color represents the adjusted *p* values. (**D**) Pathway enrichment analysis in the KEGG database. Pathways significantly enriched by DEGs are ordered by adjusted *p* values. Abbreviations: DEGs, differentially expressed genes; GO, Gene Ontology; KEGG, Kyoto Encyclopedia of Genes and Genomes.

**Figure 4 cancers-15-03899-f004:**
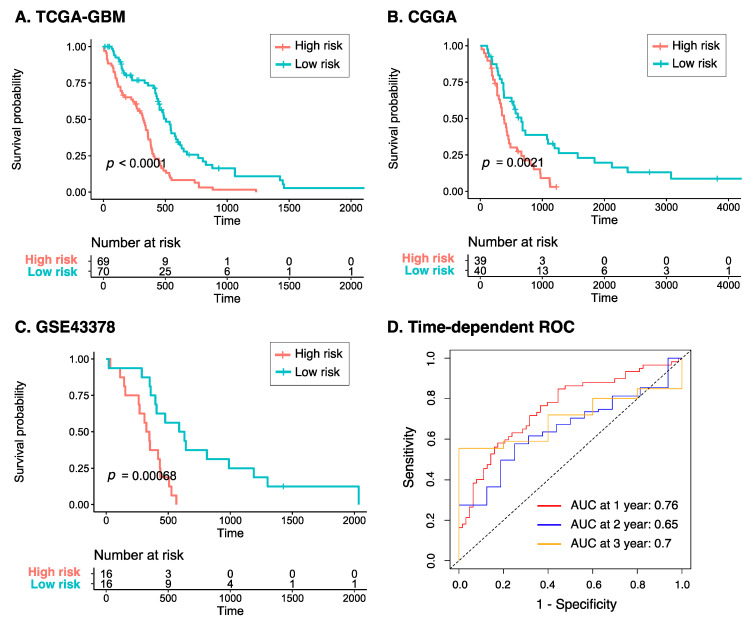
Prognostic risk score of GBM across three datasets. Kaplan–Meier plots of the eight-gene risk model for overall survival in the (**A**) TCGA-GBM, (**B**) CGGA, and (**C**) GSE43378 cohorts. (**D**) Time-dependent ROC curve analyses of the risk model at 1-, 3-, and 5-year overall survival in TCGA-GBM. Abbreviations: CGGA, Chinese Glioma Genome Atlas; ROC, receiver operating characteristic.

**Figure 5 cancers-15-03899-f005:**
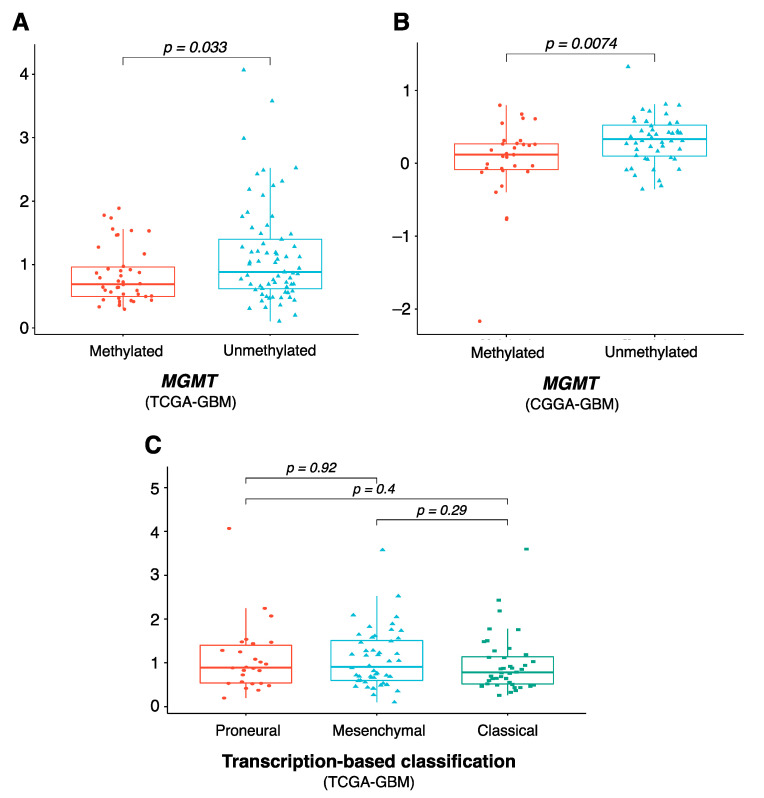
Clinical characteristics of the eight-gene risk model. Comparison of risk score level between *MGMT*-methylated and *MGMT*-unmethylated in (**A**) TCGA-GBM and (**B**) CGGA-GBM, and between (**C**) proneural, mesenchymal, and classical molecular subtypes. *p* values were obtained using the Wilcoxon test. Abbreviations: *MGMT*, O^6^-methylguanine DNA methyltransferase.

**Figure 6 cancers-15-03899-f006:**
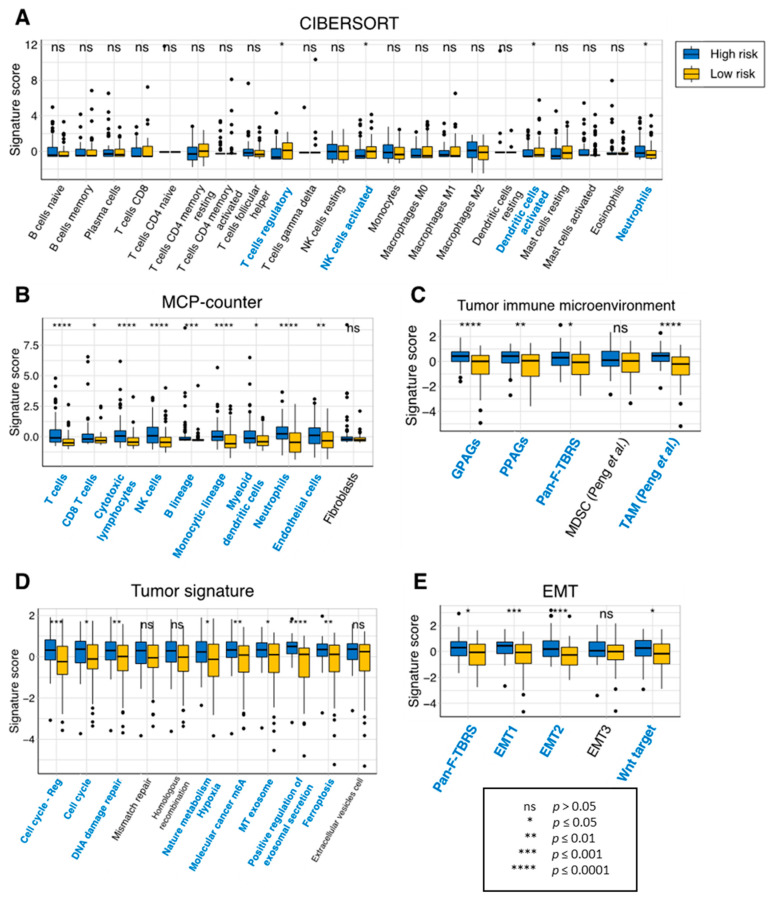
Immune infiltration of the prognostic risk score model in TCGA-GBM. Infiltration levels of immune cells between the high- and low-risk groups using the (**A**) CIBERSORT or (**B**) MCP-counter method. Comparison of measures of (**C**) the tumor immune microenvironment, (**D**) the tumor signature and (**E**) EMT between the high- and low-risk groups. Immune cells which had significantly different signature scores between the two risk groups are colored in blue. *p* values were obtained using the Wilcoxon test. Abbreviations: EMT, epithelial–mesenchymal transition; GPAGs, good-prognosis angiogenesis genes; MCP-counter, microenvironment cell populations-counter; MDSC, myeloid-derived suppressor cells; NK, natural killer; ns, not significant; Pan-F-TBRS, Pan-fibroblast TGF-β response signature; PPAGs, poor-prognosis angiogenesis genes; TAM, tumor-associated macrophages.

**Table 1 cancers-15-03899-t001:** Characteristics of the included datasets for model construction.

Dataset	Sample Size	Clinical Information	Platform	Application
TCGA-GBM	142 GBM	Female 50 (35.2%) Age 61.6 ± 11.9 (range 24–89)	Illumina	Differential expression analysis, risk score construction
GTEx (BA9)	209 normal frontal cortex samples	Female 56 (26.8%) Age group 60–70 (57.4%), 50–59 (30.1%), 40–49 (8.1%), 30–39 (2%), 20–29 (2.4%)	Illumina TruSeq	Differential expression analysis
GSE4290	81 GBM vs. 23 non-tumor (epilepsy)	Not available	Affymetrix HG-U133Plus2	Differential expression analysis
GSE68848	228 GBM vs. 28 non-tumor	Not available	Affymetrix HG-U133Plus2	Differential expression analysis
CGGA	79 GBM	Not applicable	Not applicable	Risk score validation
GSE43378	32 GBM	Not applicable	Not applicable	Risk score validation

**Table 2 cancers-15-03899-t002:** Univariate Cox regression analysis of DEGs with overall survival in the TCGA-GBM dataset.

Gene	Coefficient	Hazard Ratio	95% Confidence Interval	*p* Value
*CRNDE*	0.00047	1.00047	1.00021–1.000739	0.0004
*NRXN3*	0.00067	1.00067	1.000294–1.001064	0.0005
*POPDC3*	0.00202	1.00202	1.000876–1.003172	0.0005
*PTPRN*	0.00017	1.00017	1.000106–1.000246	8.8 × 10^−7^
*PTPRN2*	0.00012	1.00012	1.000054–1.000205	0.0007
*SLC46A2*	0.06594	1.06816	1.037777–1.098549	2.1 × 10^−5^
*TIMP1*	9.03 × 10^−6^	1.000009	1.000004–1.000014	0.0004
*TNFSF9*	0.00361	1.00362	1.001559–1.005691	0.0005

## Data Availability

Publicly available datasets were analyzed in this study and can be found at: https://www.cancer.gov/tcga, https://www.ncbi.nlm.nih.gov/geo/, http://www.cgga.org.cn/, https://gtexportal.org/ (accessed on 1 July 2023). Data generated by the authors are shown in this paper or in the Appendix A. Further data are available upon request from the corresponding author if they were not shown elsewhere.

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
