# Peer review of "Identification of a Novel Eight-Gene Risk Model for Predicting Survival in Glioblastoma: A Comprehensive Bioinformatic Analysis"

_cancers, 2023, doi:10.3390/cancers15153899_

Round 1

Reviewer 1 Report

The authors presented a study that starts from a rather interesting (although very little useful from a clinical point of view) idea such as that of developing a predictive model for glioblastoma based on the gene expression signature. As they themselves mentioned in the introduction, the idea is not entirely original and it is not clear how this work overcomes the problems and criticalities of other approaches, improving the results.

The whole first part of the study based on the identification of the DEGs (which is not clear how it is connected to the rest of the results) needs to be deeply reviewed given the enormous imbalance in the number of GBM and control samples in two out of three of the datasets analyzed (I strongly suggest using GTEx as the source of the normal checks for GBM-TCGA data). Furthermore, for the DE analysis it could be useful to include factors such as age, gender or therapies followed in the DESeq2 and limma designs.

In the second part of the work the authors focus on the identification of genes associated with the prognosis of GBM (the previous analysis was not necessary to this aim), but it is not clear to me why the authors focused only on the TCGA dataset to apply the univariate Cox regression since all the three considered datasets contain both gene expression and survival data. It is also unclear how the risk score was calculated. I suggest testing if the genes obtained are somehow interconnected with a network analysis.

The third and last part of the work sound a bit like a sort of "flight of fancy" and both the methods and the results are described in a very superficial way.

MINOR ISSUES: the figures are quite badly presented and there are several errors in the graphics; in figure 1 there is nothing in the volcano plots; in figures 4 and 5 it is not reported which statistical test was used to calculate the p-values; figure 6 is not mentioned in the text.

The quality of the English is generally quite good and does not need particularly extensive corrections.

Author Response

cancers-2467678 (July 10 2023)

REVIEWER #1 (Remarks to the Authors):

The authors presented a study that starts from a rather interesting (although very little useful from a clinical point of view) idea such as that of developing a predictive model for glioblastoma based on the gene expression signature. As they themselves mentioned in the introduction, the idea is not entirely original and it is not clear how this work overcomes the problems and criticalities of other approaches, improving the results.

R1.1. We thank the reviewer for their comment. In response to the reviewer's fair skepticism about the study's clinical utility, we would argue the following points. Despite undergoing aggressive treatments, GBM patients have poor overall survival and a high rate of recurrence, warranting efforts to look for more effective therapies. This has been somewhat achievable with previous findings of the significance of MGMT promoter methylation status. We used publicly available datasets to inquire patterns of genetic alterations that could influence prognosis of GBM, which can serve as a basis for subsequent studies that can confirm its significance. This approach understandably has its strengths and limitations, as we outlined in the Discussion. Bioinformatics as a field has truly boomed in the past decade and we acknowledge that a large part of our study workflow is similar to that of previous studies. What we hope to achieve was to extend these approaches, and we have rewritten relevant parts of the Introduction (line 69-71) and Conclusion (line 349-352) to clarify this point.

The whole first part of the study based on the identification of the DEGs (which is not clear how it is connected to the rest of the results) needs to be deeply reviewed given the enormous imbalance in the number of GBM and control samples in two out of three of the datasets analyzed (I strongly suggest using GTEx as the source of the normal checks for GBM-TCGA data). Furthermore, for the DE analysis it could be useful to include factors such as age, gender or therapies followed in the DESeq2 and limma designs.

R1.2. We thank the reviewer for their very detailed feedback.

The concept of DEGs stems from the availability of RNA-sequencing, which provides a great amount of information of gene expression levels, followed by identification of genes that are significantly differentially expressed across two or more conditions, in this context GBM and normal controls. It is now one of the most common steps in analyzing RNA-seq data (McDermaid et al. 2019). Our study aims to look for prognostic markers of GBM, and so we began with identification of DEGs and then see if they were truly associated with a better or poorer survival in GBM. A different approach would be to begin with a hypothesis about a gene or set of genes of interest, for example whether it is related to a pathway known to drive cancers, as was performed in our previous publications (H. H. Dang et al. 2022; H.-H. Dang et al. 2021). Regardless, an important step is to ensure that the genes as mentioned are expressed differently between tumor and normal samples, before inquiring their prognostic potential. Hence, the identification of the DEGs serves as the stepping stone of our study, and has to be performed followed by the remaining analyses (eg enrichment pathway analysis, construction of risk score, etc).

To address the reviewer's concern about the imbalance in sample size between GBM cases and controls, we have:

  1. Replaced the dataset GSE7696, which included only 4 non-tumor samples, with GSE68848 of the REMBRANDT (Repository of Molecular Brain Neoplasia Data) study. The description is at line 83-86.

  1. Included control samples from the Genotype-Tissue Expression (GTEx) database (n = 209) for differential expression analysis against GBM samples from the TCGA dataset.

The inclusion of age, sex or therapies in the DEG analysis as the reviewer suggested could have clinical implications, but we have opted not to do so for the following reasons:

  1. Almost all of the GBM patients in the datasets included in this study were untreated (90% of cases in TCGA-GBM were newly diagnosed (McLendon et al. 2008)). Another reason we replaced GSE7696 with GSE68848, other than that it contained few non-tumor samples as mentioned above, was that GBM patients in GSE7696 were treated in clinical trials, which would make comparisons between this cohort and control samples inappropriate to be combined with other GEO datasets that contained mainly untreated patients.

  1. It is widely known that men are more likely to develop GBM than women and that it is generally diagnosed at older age (Tamimi and Juweid 2017; Yang et al. 2019).

  1. Inclusion of age, sex and therapies is not common as part of the DEG analytical workflow (Love, Huber, and Anders 2014).

In the second part of the work the authors focus on the identification of genes associated with the prognosis of GBM (the previous analysis was not necessary to this aim), but it is not clear to me why the authors focused only on the TCGA dataset to apply the univariate Cox regression since all the three considered datasets contain both gene expression and survival data. It is also unclear how the risk score was calculated. I suggest testing if the genes obtained are somehow interconnected with a network analysis.

R1.3. We have argued the case about DEGs in response to the reviewer's previous comment, and hope that it will also serves us here.

We would like to clarify that survival data for GSE4290 and GSE68848 were not available (even though they were presented in the publications associated with these datasets). Specifically, we used the R package GEOquery (Davis and Meltzer 2007) and patient phenotype data did not contain survival information. Hence only data from TCGA-GBM were used to perform univariate Cox regression analysis.

Regarding the calculation of risk score for each patient, we have added a formula at line 219-221.

The network analysis was also performed and mentioned at line 215-216.

The third and last part of the work sound a bit like a sort of "flight of fancy" and both the methods and the results are described in a very superficial way.

R1.4. We believe the third part here refers to sections 2.3. and 3.2.. We have added relevant parts for better clarification (Section 2.3 at line 117-125, Section 3.2 at line 193-201). Figure 2 relating to this part was also improved for better visualization (described below in response to the reviewer's final comment).

Regarding the last part, we acknowledge that the relationship between the 8-gene signature with disease heterogeneity as mapped by the cell-state plot remains subjective and largely qualitative in nature, and therefore have decided to remove this part from our study.

MINOR ISSUES: the figures are quite badly presented and there are several errors in the graphics; in figure 1 there is nothing in the volcano plots; in figures 4 and 5 it is not reported which statistical test was used to calculate the p-values; figure 6 is not mentioned in the text.

R1.5. To address these concerns, we have modified the figures as follows:

Figure 1. The data have now been added. We apologized that it failed to show when we converted the manuscript word to PDF file and this error went unnoticed. We also added the data legend to clarify if a gene was up- or downregulated, or not significant.

Figure 2. The size of pathway names have been enlarged. P values of GO analysis have been rounded to 1 decimal place for brevity.

Figure 3. A legend for high and low risk group has been added.

Figure 4. The statistical test to calculate p values (Wilcoxon test) was added in the figure caption. 

Figure 5. Immune cells with significant infiltration levels and measures were highlight in blue. The color label of Figure 5A has been changed to blue and yellow for consistency with the remaining parts. Only one data legend remained on the top right corner for all sub-figures. Also, the statistical test to calculate p values (Wilcoxon test) was added in the figure caption. 

Figure 6 was removed as mentioned above.

Additionally, in all figures the axis titles and value labels have been enlarged for better visualization.

REFERENCES

Dang, Huy Hoang, Hoang Dang Khoa Ta, Truc T. T. Nguyen, Gangga Anuraga, Chih-Yang Wang, Kuen-Haur Lee, and Nguyen Quoc Khanh Le. 2022. “Prospective Role and Immunotherapeutic Targets of Sideroflexin Protein Family in Lung Adenocarcinoma: Evidence from Bioinformatics Validation.” Functional & Integrative Genomics 22 (5): 1057–72. https://doi.org/10.1007/s10142-022-00883-3.

Dang, Huy-Hoang, Hoang Dang Khoa Ta, Truc T. T. Nguyen, Gangga Anuraga, Chih-Yang Wang, Kuen-Haur Lee, and Nguyen Quoc Khanh Le. 2021. “Identifying GPSM Family Members as Potential Biomarkers in Breast Cancer: A Comprehensive Bioinformatics Analysis.” Biomedicines 9 (9): 1144. https://doi.org/10.3390/biomedicines9091144.

Davis, Sean, and Paul S. Meltzer. 2007. “GEOquery: A Bridge between the Gene Expression Omnibus (GEO) and BioConductor.” Bioinformatics (Oxford, England) 23 (14): 1846–47. https://doi.org/10.1093/bioinformatics/btm254.

Love, Michael I., Wolfgang Huber, and Simon Anders. 2014. “Moderated Estimation of Fold Change and Dispersion for RNA-Seq Data with DESeq2.” Genome Biology 15 (12): 550. https://doi.org/10.1186/s13059-014-0550-8.

McDermaid, Adam, Brandon Monier, Jing Zhao, Bingqiang Liu, and Qin Ma. 2019. “Interpretation of Differential Gene Expression Results of RNA-Seq Data: Review and Integration.” Briefings in Bioinformatics 20 (6): 2044–54. https://doi.org/10.1093/bib/bby067.

McLendon, Roger, Allan Friedman, Darrell Bigner, Erwin G. Van Meir, Daniel J. Brat, Gena M. Mastrogianakis, Jeffrey J. Olson, et al. 2008. “Comprehensive Genomic Characterization Defines Human Glioblastoma Genes and Core Pathways.” Nature 455 (7216): 1061–68. https://doi.org/10.1038/nature07385.

Tamimi, Ahmad Faleh, and Malik Juweid. 2017. “Epidemiology and Outcome of Glioblastoma.” In Glioblastoma, edited by Steven De Vleeschouwer. Brisbane (AU): Codon Publications. http://www.ncbi.nlm.nih.gov/books/NBK470003/.

Yang, Wei, Nicole M. Warrington, Sara J. Taylor, Paula Whitmire, Eduardo Carrasco, Kyle W. Singleton, Ningying Wu, et al. 2019. “Sex Differences in GBM Revealed by Analysis of Patient Imaging, Transcriptome, and Survival Data.” Science Translational Medicine 11 (473): eaao5253. https://doi.org/10.1126/scitranslmed.aao5253.

Reviewer 2 Report

This study identified 13 genes that were differentially regulated in Glioblastoma. The main problem I see with the approach used to identify these genes was that they combined microarray data from tumors of different types (glioma, astrocytoma and oligodendroglioma for dataset GSE4790; tumor samples from patients on clinical trials in GSE7696 and TCGA datasets) which have different properties. This may result in inappropriate comparisons. The number of control samples (admittedly hard to obtain) is much smaller than the number of GBM samples, and include a significan number of epilepsy patient samples, which are non-tumor samples from diseased brains. Figure 1  (Summary of DEGs) indicates that volcano plots would be included, but volcano plots were missing, only the axis were present. The DEGs associated gene lists were not included either, with the fold change and p-values obtained, the lists should have been included in Supplementary information. For the characterization of the risk model based subgroups, the BRAF status comparison only appears to have one sample in the BRAF mutant subgroup, the rest of the samples had a wild type genotype (Figure 4 B), hence, the comparison groups appear be inadequate. 

Author Response

cancers-2467678 (July 10 2023)

REVIEWER #2 (Remarks to the Authors):

This study identified 13 genes that were differentially regulated in Glioblastoma. The main problem I see with the approach used to identify these genes was that they combined microarray data from tumors of different types (glioma, astrocytoma and oligodendroglioma for dataset GSE4290; tumor samples from patients on clinical trials in GSE7696 and TCGA datasets) which have different properties. This may result in inappropriate comparisons. The number of control samples (admittedly hard to obtain) is much smaller than the number of GBM samples, and include a significant number of epilepsy patient samples, which are non-tumor samples from diseased brains.

R2.1. We thank the reviewer for their detailed feedback. To address their concern, we have:

  1. Replaced the dataset GSE7696, which included patients who had recurrent disease, with GSE68848 of the REMBRANDT (Repository of Molecular Brain Neoplasia Data) study. The description is at line 83-86.

  1. Included control samples from the Genotype-Tissue Expression (GTEx) database (n = 209) for differential expression analysis against GBM samples from the TCGA dataset.

We would like to clarify that only data of GBM patients were included in our study. Data of other disease types (astrocytoma and oligodendroglioma in GSE4290; oligodendroglioma, astrocytoma, mixed, unclassified and unknown in GSE68848) were excluded. We have included this descriptions at line 82-86.

Regarding TCGA dataset, we decided to include it for two reasons. Firstly, the proportions of newly diagnosed (i.e. untreated) / secondary / recurrent GBM cases in the TCGA dataset were 90% / 2% / 8%, respectively, hence the majority of samples from TCGA are still primary GBM (McLendon et al. 2008). Secondly, GBM was the first cancer to be systematically studied by TCGA and the dataset TCGA-GBM has been used extensively and generated much of the knowledge that we know so far about GBM (Brennan et al. 2013).

Regarding the brain samples of epilepsy patients as non-tumor controls in GSE4290, we acknowledge your concern. The original study that generated GSE4290 was published in 2006 and understandably data from the so-called control group was collected in a way somewhat akin to "convenience sampling" (as stated by the authors, "... nontumor brain tissues of epilepsy were obtained from surgery patients...") (Sun et al. 2006). Nevertheless, the reviewer also mentioned that control samples are, admittedly, hard to obtain, and we believe that the inclusion of GSE4290 would not invalidate the results substantially.

Figure 1 (Summary of DEGs) indicates that volcano plots would be included, but volcano plots were missing, only the axis were present.

R2.2. The data for figure 1 have now been added. We apologized that it failed to show when we converted the manuscript word to PDF file and this error went unnoticed.

The DEGs associated gene lists were not included either, with the fold change and p-values obtained, the lists should have been included in Supplementary information.

R2.3. Thank you for your suggestion. The lists of DEGs have been added as Supplementary Table S1.

For the characterization of the risk model based subgroups, the BRAF status comparison only appears to have one sample in the BRAF mutant subgroup, the rest of the samples had a wild type genotype (Figure 4 B), hence, the comparison groups appear be inadequate.

R2.4. Thank you for your suggestion. We have decided that it would be better to remove the comparison of BRAF statuses, considering that BRAF alterations are rare in GBM (Schindler et al. 2011). Instead we have added the analysis of MGMT status using data from GBM patients in CGGA cohort (Figure 4B). We also tried to perform this analysis with EGFR amplification and CDKN2A deletion, which would be more clinically relevant, however such information was unfortunately not available in the datasets used in this study.

REFERENCES

Brennan, Cameron W., Roel G. W. Verhaak, Aaron McKenna, Benito Campos, Houtan Noushmehr, Sofie R. Salama, Siyuan Zheng, et al. 2013. “The Somatic Genomic Landscape of Glioblastoma.” Cell 155 (2): 462–77. https://doi.org/10.1016/j.cell.2013.09.034.

McLendon, Roger, Allan Friedman, Darrell Bigner, Erwin G. Van Meir, Daniel J. Brat, Gena M. Mastrogianakis, Jeffrey J. Olson, et al. 2008. “Comprehensive Genomic Characterization Defines Human Glioblastoma Genes and Core Pathways.” Nature 455 (7216): 1061–68. https://doi.org/10.1038/nature07385.

Schindler, Genevieve, David Capper, Jochen Meyer, Wibke Janzarik, Heymut Omran, Christel Herold-Mende, Kirsten Schmieder, et al. 2011. “Analysis of BRAF V600E Mutation in 1,320 Nervous System Tumors Reveals High Mutation Frequencies in Pleomorphic Xanthoastrocytoma, Ganglioglioma and Extra-Cerebellar Pilocytic Astrocytoma.” Acta Neuropathologica 121 (3): 397–405. https://doi.org/10.1007/s00401-011-0802-6.

Sun, Lixin, Ai-Min Hui, Qin Su, Alexander Vortmeyer, Yuri Kotliarov, Sandra Pastorino, Antonino Passaniti, et al. 2006. “Neuronal and Glioma-Derived Stem Cell Factor Induces Angiogenesis within the Brain.” Cancer Cell 9 (4): 287–300. https://doi.org/10.1016/j.ccr.2006.03.003.

Reviewer 3 Report

The paper entitled Identification of a Novel 13-gene Risk Model for Predicting Survival in Glioblastoma: A Comprehensive Bioinformatic Analysis, presented by Huy-Hoang Dang et al, develops a 13-gene signature to calculate a risk score for death in glioblastoma patients. The work is robust and uses diverse data sets from different samples, which minimizes the inclusion of possible biases in the final result. Likewise, the methodology carried out is coherent and reasonable with the objectives of the study. On the other hand, although the approach of developing a glioblastoma risk score from transcriptomic data is not novel, the authors study the relationship of the proposed score in the context of GBM heterogeneity, showing that the model they propose has a certain independence with the different GBM subtypes.  For all the above I would like to congratulate the authors for the work presented. My recommendation is to publish the work, but previously some major and minor issues should be addressed, which I expose below.

Major issues:

In the 2021 fifth edition of the WHO classification of CNS tumors, it is specified that glioblastomas will be IDH wt (10.3174/ajnr.A7462). However, previous cohorts do show IDH mt GBMs, whose prognosis is clearly superior to that of IDH wt GBMs. Therefore, since the presence of IDH mt may introduce a clear bias in survival and since IDH mt patients will not be considered GBM in the future, it is necessary to repeat the analysis filtering out IDH wt samples. This will favor the future clinical applicability of the proposed score.

BRAF is a rare alteration in glioblastoma, as indicated in the reference 24 that you cite in the paper. In fact, Figure 4B shows that only 2 patients in the entire TCGA cohort present this mutation. I believe that it would be more appropriate to analyze other more common alterations in GBM such as EGFR amplification or CDKN2A deletion.

In Figure 3B the high survival of the low-risk group has caught my attention. In addition, the number of samples is greater than the total number of GBM in the CGGA, so it would be necessary to check that a low-grade glioma has not been introduced by mistake in the analysis.

Minor issues:

Lines 144-145 may lead to confusion. As well explained in the lines immediately above, GSVA is a method to assess the enrichment of a set of genes in a sample. But ssGSEA is a method similar to GSVA, which pursues the same thing and was developed independently, but is not an extension of GSVA. It is true that there is an R package also called GSVA which, in addition to the method with the same name, also implements ssGSEA. The wording of these two lines should be revised. In addition, the reference of the paper where the ssGSEA methodology is presented should be added (DOI: 10.1038/nature08460).

Section 3.2 focuses the presentation of results on the extracellular matrix; however, there is also enrichment in pathways related to the immune system (Figure 2). It seems appropriate to highlight these results here as they are related to the results shown later in Figure 5.

I suggest the following changes in tables and figures:

- In Table 1 I would add a column indicating the platform used to measure RNA expression (RNA-seq, Affymetrix, etc.). I would also add in this table the datasets that will be used later in the validation of the score.

- Figure 1 does not show the data in the graph.

- In Table 2 I recommend replacing the standard error with the 95% CI of the HR, as this would facilitate the interpretation of the results.

- It has been seen that the neural subtype corresponds to samples with a high content of non-tumor tissue rather than a GBM subtype per se (10.1016/j.ccell.2017.06.003, 10.1093/bib/bbaa129), therefore I recommend that you use the three-subtype model: proneural, classical and mesenchymal (10.1016/j.ccell.2017.06.003) instead of the 4-subtype model.

- In Figure 6 there is a typo in the upper right label, it should be NPC-like.

I would also like to point out the following errata or aspects of the wording that should be clarified:

- In section 2.2 it would be appropriate to specify that the differential expression analysis was carried out by pitting the GBM samples against the control samples of each dataset.

- Line 117 indicates that the threshold of 0.0001 will be used to consider a significant result. In order to assess the appropriateness of this threshold it would be necessary to know the number of genes to be tested or, failing that, to apply some p-value adjustment algorithm to control for type I error.

- In line 174 it is indicated that, for GEO datasets, it was filtered with log FC > 1. However, in line 103 it is said that the filter was log FC > 0.5. The appropriate one should be corrected.

I hope that these comments will contribute to the improvement of this work.

Best regards

Author Response

cancers-2467678 (July 8 2023)

REVIEWER #3 (Remarks to the Authors):

The paper entitled Identification of a Novel 13-gene Risk Model for Predicting Survival in Glioblastoma: A Comprehensive Bioinformatic Analysis, presented by Huy-Hoang Dang et al, develops a 13-gene signature to calculate a risk score for death in glioblastoma patients. The work is robust and uses diverse data sets from different samples, which minimizes the inclusion of possible biases in the final result. Likewise, the methodology carried out is coherent and reasonable with the objectives of the study. On the other hand, although the approach of developing a glioblastoma risk score from transcriptomic data is not novel, the authors study the relationship of the proposed score in the context of GBM heterogeneity, showing that the model they propose has a certain independence with the different GBM subtypes.  For all the above I would like to congratulate the authors for the work presented. My recommendation is to publish the work, but previously some major and minor issues should be addressed, which I expose below.

We are thankful to the reviewer for their very detailed feedback on the manuscript. Addressing their concerns has significantly improved the manuscript, and substantiated the validity of the results.

Major issues:

In the 2021 fifth edition of the WHO classification of CNS tumors, it is specified that glioblastomas will be IDH wt (10.3174/ajnr.A7462). However, previous cohorts do show IDH mt GBMs, whose prognosis is clearly superior to that of IDH wt GBMs. Therefore, since the presence of IDH mt may introduce a clear bias in survival and since IDH mt patients will not be considered GBM in the future, it is necessary to repeat the analysis filtering out IDH wt samples. This will favor the future clinical applicability of the proposed score.

R3.1. We have revised the analysis using data from only IDH-wildtype patients in the TCGA-GBM dataset (Methods, line 86-90 and 126-127). While the number of GBM samples was slightly smaller (142 only IDH-wildtype cases vs. 161 both IDH-wildtype and -mutant cases) and hence the change of DEGs, we continued to see significant results in the construction of the risk model. This revision, however, was not possible with other datasets as they did not contain informtion on IDH status of GBM cases. We have added these points to the Discussion as one of the study's limitations, lines 331-342.

BRAF is a rare alteration in glioblastoma, as indicated in the reference 24 that you cite in the paper. In fact, Figure 4B shows that only 2 patients in the entire TCGA cohort present this mutation. I believe that it would be more appropriate to analyze other more common alterations in GBM such as EGFR amplification or CDKN2A deletion.

R3.2. Thank you for this suggestion. We have decided that it would be better to remove the comparison of BRAF statuses, instead we have added the analysis of MGMT status using data from GBM patients in CGGA cohort (Figure 4B). We agree that results with EGFR amplification and CDKN2A deletion would be more clinically relevant, however such information was unfortunately not available in the datasets used in this study.

In Figure 3B the high survival of the low-risk group has caught my attention. In addition, the number of samples is greater than the total number of GBM in the CGGA, so it would be necessary to check that a low-grade glioma has not been introduced by mistake in the analysis.

R3.3. We acknowledge that we incorrectly included data of low-grade glioma cases in the original version of the manuscript as the reviewer pointed out. The revised results now have only patients with GBM in the CGGA cohort (sample size n = 79, compared to n = 313 in the original manuscript).

Minor issues:

Lines 144-145 may lead to confusion. As well explained in the lines immediately above, GSVA is a method to assess the enrichment of a set of genes in a sample. But ssGSEA is a method similar to GSVA, which pursues the same thing and was developed independently, but is not an extension of GSVA. It is true that there is an R package also called GSVA which, in addition to the method with the same name, also implements ssGSEA. The wording of these two lines should be revised. In addition, the reference of the paper where the ssGSEA methodology is presented should be added (DOI: 10.1038/nature08460).

R3.4. We are grateful for the reviewer for this correction. We have rewritten this part as follows: "Specifically, we employed the GSVA R package GSVA, implementing the single-sample gene set enrichment analysis (ssGSEA) method" (line 156-157) and added the reference Barbie et al. Nature 2009.

Section 3.2 focuses the presentation of results on the extracellular matrix; however, there is also enrichment in pathways related to the immune system (Figure 2). It seems appropriate to highlight these results here as they are related to the results shown later in Figure 5.

R3.5. We thank you for the thorough reading. To address another reviewer's concern that a dataset used for differential gene expression (DEG) analysis contained data from treated GBM patients (GSE7696), our revision has replaced it with GSE68848. Hence, the DEGs and subsequent results have changed, with GO and KEGG analysis now shows enrichment in pathways mostly related to the synapse.

I suggest the following changes in tables and figures:

- In Table 1 I would add a column indicating the platform used to measure RNA expression (RNA-seq, Affymetrix, etc.). I would also add in this table the datasets that will be used later in the validation of the score.

R3.6. We have now added these information in Table 1.

- Figure 1 does not show the data in the graph.

R3.7. The data for figure 1 have now been added. We apologized that it failed to show when we converted the manuscript word to PDF file and this error went unnoticed.

- In Table 2 I recommend replacing the standard error with the 95% CI of the HR, as this would facilitate the interpretation of the results.

R3.8. We have revised Table 2 as you recommended.

- It has been seen that the neural subtype corresponds to samples with a high content of non-tumor tissue rather than a GBM subtype per se (10.1016/j.ccell.2017.06.003, 10.1093/bib/bbaa129), therefore I recommend that you use the three-subtype model: proneural, classical and mesenchymal (10.1016/j.ccell.2017.06.003) instead of the 4-subtype model.

R3.9. We thank you for this suggestion. The 3-subtype model is now used in place of the 4-subtype model. We have also mentioned this information in the Methods (line 149-152) and Results (line 250-252), and added the references.

- In Figure 6 there is a typo in the upper right label, it should be NPC-like.

R3.10. We thank the reviewer for the thorough reading. However, in light of another reviewer's concern about the validity of this part of the result, we felt that the relationship between the 8-gene signature with disease heterogeneity as mapped by the cell-state plot remains subjective and largely qualitative in nature, and therefore have decided to remove it from our study. We understand that this would somewhat reduce the novelty of our study, and have rewritten relevants part throughout the manuscript to reflect this.

I would also like to point out the following errata or aspects of the wording that should be clarified:

- In section 2.2 it would be appropriate to specify that the differential expression analysis was carried out by pitting the GBM samples against the control samples of each dataset.

R3.11. Thank you for this suggestion. We have added this description in the Methods (section 2.2., line 104-105) and Figure 1 caption (line 180-181).

- Line 117 indicates that the threshold of 0.0001 will be used to consider a significant result. In order to assess the appropriateness of this threshold it would be necessary to know the number of genes to be tested or, failing that, to apply some p-value adjustment algorithm to control for type I error.

R3.12. We have added a Supplementary Table S2 to show statistical values of the Cox regression analysis on all 1,934 DEGs. We would also like to correct a typo, which was that the p value threshold was 0.001, not 0.0001, and we apologize for this error.

- In line 174 it is indicated that, for GEO datasets, it was filtered with log FC > 1. However, in line 103 it is said that the filter was log FC > 0.5. The appropriate one should be corrected.

R3.13. Thank you for your thorough reading. We decided to change the cutoff value of log Fold change to be greater than 1 for all DEG analysis between GBM and control samples for consistency (previously it was > 0.5 for GEO datasets). We have stated it in the Methods (section 2.2, line 108-111) and Results (section 3.1., line 171-172)

I hope that these comments will contribute to the improvement of this work.

Best regards

We are truly grateful for your extremely constructive feedback on our manuscript. We hope the revised version has now substantially addressed your comments.

Round 2

Reviewer 1 Report

The quality of the manuscript has generally improved but there are still some problems: first of all, if there are no errors in the summary table of the sources/types datasets used, I really don't understand why the authors used TCGA-GBM data of gene expression from Affymetrix instead of using those from RNA-Seq in order to avoid any possible bias due to the different technologies compared to the control brain GTEx. This point must absolutely be solved in a mandatory way. Of course I'm not asking for the raw data (fastq) to be reprocessed but only to download the raw gene expression counts (absolutely not the TPMs!) in order to standardize the analysis with the GTEx data. 

an important step is to ensure that the genes as mentioned are expressed differently between tumor and normal samples, before inquiring their prognostic potential”  I do not agree 100% because theoretically it is possible for each gene (regardless of whether it is down/up regulated in GBM vs normal brain) to stratify patients into low and high expression for example using the median as a cut-off in in order to obtain two groups of comparable size and using Kaplan-Meier survival curves test whether the high or low expression is significantly associated with a better or worse prognosis of the patients.

Regarding the Functional enrichment analysis of DEGs I suggest to separate the up and down regulated genes and produce two separate sets of graphs otherwise the transmitted message is useless.

Inclusion of age, sex and therapies is not common as part of the DEG analytical workflow I certainly agree with the authors on this point (which is why my request was not mandatory but only a suggestion) on the fact that normally this type of corrections are not inserted in the DESeq2 design but in the case of a dataset like this is inherently biased (as you you say "It is widely known that men are more likely to develop GBM than women" and "Almost all of the GBM patients in the datasets included in this study were untreated") correcting this effect could help to give a result that is valid for the GBM and no longer for the Men's GBM; however, I repeat that it was only a suggestion.

Author Response

cancers-2467678 (July 26 2023)

REVIEWER #1 (Remarks to the Authors):

We are thankful to the reviewer for their very detailed feedback on the manuscript. Addressing their concerns has significantly improved the manuscript, and substantiated the validity of the results.

The quality of the manuscript has generally improved but there are still some problems: first of all, if there are no errors in the summary table of the sources/types datasets used, I really don't understand why the authors used TCGA-GBM data of gene expression from Affymetrix instead of using those from RNA-Seq in order to avoid any possible bias due to the different technologies compared to the control brain GTEx. This point must absolutely be solved in a mandatory way. Of course I'm not asking for the raw data (fastq) to be reprocessed but only to download the raw gene expression counts (absolutely not the TPMs!) in order to standardize the analysis with the GTEx data.

Thank you very much for your careful review. We apologize for our error in the original manuscript. We have corrected "Affymetrix HG-U133A" to "Illumina" as the platform of TCGA-GBM gene expression data in Table 1.

(Reference: https://xenabrowser.net/datapages/?dataset=TCGA-GBM.htseq_counts.tsv&host=https%3A%2F%2Fgdc.xenahubs.net&removeHub=https%3A%2F%2Fxena.treehouse.gi.ucsc.edu%3A443)

“an important step is to ensure that the genes as mentioned are expressed differently between tumor and normal samples, before inquiring their prognostic potential”  I do not agree 100% because theoretically it is possible for each gene (regardless of whether it is down/up regulated in GBM vs normal brain) to stratify patients into low and high expression for example using the median as a cut-off in in order to obtain two groups of comparable size and using Kaplan-Meier survival curves test whether the high or low expression is significantly associated with a better or worse prognosis of the patients.

We appreciate your insightful comment. We have reviewed the literature and added a paragraph about the limitation of DEGs analysis in the Discussion, read as follows (line 315-323):

"Our study aims to determine prognostic markers of GBM, with the identification of DEGs followed by survival analysis. The concept of DEGs stems from the availability of expression of mRNA transcripts, which allows identification of genes that are significantly differentially expressed across two or more conditions, in this context GBM and normal controls. It is one of the most common steps in analyzing microarray and RNA-seq data [37]. However, critics have argued that while DEGs can help serve as clinical biomarkers  or obtain mechanistic insights into diseases, they do not necessarily represent causes between gene expression and phenotype, but could be consequences or simply correlations [38]. [...] "

Regarding the Functional enrichment analysis of DEGs I suggest to separate the up and down regulated genes and produce two separate sets of graphs otherwise the transmitted message is useless.

Thank you very much for the suggestion. The functional enrichment analysis has now been performed separately for DEGs that were consistently upregulated (939 DEGs) or downregulated across the three comparisons (733 DEGs). The results are shown in Figures 2 and 3.

Inclusion of age, sex and therapies is not common as part of the DEG analytical workflow I certainly agree with the authors on this point (which is why my request was not mandatory but only a suggestion) on the fact that normally this type of corrections are not inserted in the DESeq2 design but in the case of a dataset like this is inherently biased (as you you say "It is widely known that men are more likely to develop GBM than women" and "Almost all of the GBM patients in the datasets included in this study were untreated") correcting this effect could help to give a result that is valid for the GBM and no longer for the Men's GBM; however, I repeat that it was only a suggestion.

We sincerely appreciate your correction and comments. We agree that the inclusion of age, sex and therapies in the DEG analytical workflow would indeed provide a valuable result to the study, and it certainly has been done before (Porcu et al., Nat Commun 2021; Brooks et al., Front Neurosci 2019). We have reviewed the datasets and included the clinical information of participants from TCGA-GBM and GTEx in Table 1. Clinical data of those from GEO datasets were unfortunately not available. We have also added a paragraph in the Discussion to raise this issue as a limitation of our study (line 320-327):

" [...] However, critics have argued that while DEGs can help serve as clinical biomarkers  or obtain mechanistic insights into diseases, they do not necessarily represent causes between gene expression and phenotype, but could be consequences or simply correlations [38]. There are a few approaches to circumvent this issue. For instance, baseline characteristics of participants, such as age, sex, and treatment status, are known to be a significant source of variations in differential expression testing [38,39]. Incorporating them into the DEG analytical workflow was unfortunately not possible in our study due to the lack of clinical data in the GEO datasets. [...] "

Reviewer 2 Report

The revised manuscript still is lacking many of the things listed in the first review.

For example, Supplementary table 1 has the fold changes and p-values of the compared datasets, but it should have the fold changes and p-values for each type of comparison for the DEGs in common. This would allow the readers to compare the consistency of DEGs across the datasets. The authors indicate that the concept of DEGs arose for RNA Seq datasets, when it had been used for a long time for expression analysis of microarray data. Indeed, the majority of the data used for the secondary analyses in this manuscript came from expression microarray datasets. I agree with Reviewer  #1 that this manuscript needs major revision, most notably the lack of validation of their results, apart from the survival data and the immunological analysis presented.

For example, Supplementary figure S2 only indicates in red the genes proposed to be high risk bio markers, but we do not know from this figure if other genes in the network are DEGs and how are they changing between the expression datasets.

English language is not a major problem in this manuscript.

Author Response

cancers-2467678 (July 26 2023)

REVIEWER #2 (Remarks to the Authors):

We are thankful to the reviewer for their very detailed feedback on the manuscript. Addressing their concerns has significantly improved the manuscript, and substantiated the validity of the results.

The revised manuscript still is lacking many of the things listed in the first review.

For example, Supplementary table 1 has the fold changes and p-values of the compared datasets, but it should have the fold changes and p-values for each type of comparison for the DEGs in common. This would allow the readers to compare the consistency of DEGs across the datasets.

We thank the reviewer for their very helpful comment. We have now added the fold changes and p-values of all compared datasets of the common DEGs in the Supplementary Table 1, as well as whether each DEG was consistently upregulated or downregulated in all comparisons.

The authors indicate that the concept of DEGs arose for RNA Seq datasets, when it had been used for a long time for expression analysis of microarray data. Indeed, the majority of the data used for the secondary analyses in this manuscript came from expression microarray datasets. I agree with Reviewer  #1 that this manuscript needs major revision, most notably the lack of validation of their results, apart from the survival data and the immunological analysis presented.

We thank the reviewer for their critical suggestion, and apologize for this error. We have addressed this limitation of our study in the Discussion, read as follows:

"Our study aims to determine prognostic markers of GBM, with the identification of DEGs followed by survival analysis. The concept of DEGs stems from the availability of expression of mRNA transcripts, which allows identification of genes that are significantly differentially expressed across two or more conditions, in this context GBM and normal controls. It is one of the most common steps in analyzing microarray and RNA-seq data [37]." (line 315-320)

"The main limitation of the study is that data were largely procured from online databases. Part of the DEGs analysis was performed using microarray data from GEO datasets, which are known to be inferior in the quality of transcriptome profiling compared to RNA-Seq data. [...] Importantly, the preliminary nature of the findings entice further experimental validations to confirm them." (line 363-370)

For example, Supplementary figure S2 only indicates in red the genes proposed to be high risk biomarkers, but we do not know from this figure if other genes in the network are DEGs and how are they changing between the expression datasets.

We apologize for the lack of clarification. The other genes in the network (the purple ones) are transcription factors and not DEGs. We have now added the description of this analysis in the Methods and Results, read as follows:

"We then predict the upstream regulators (transcription factors, TFs) of the risk-related gene candidates using NetworkAnalyst (version 3.0, http://www.networkanalyst.ca) [29]. TF-gene interaction analysis was performed with the ChIP Enrichment Analysis (ChEA) database [30]." (Methods, line 153-156)

"Using NetworkAnalyst, we observed a TFs-gene biomarker regulatory network including 143 interaction pairs among 7 seed genes (in red) and 86 TFs (in purple) (Supplementary Figure S2). Specifically, PTPRN2 was regulated by the most TFs (38 TFs), followed by NRXN3 and TNFSF9 (27 and 21 TFs, respectively)." (Results, line 278-281)

Reviewer 3 Report

I would like to thank the authors for their efforts in responding to my comments. I believe that the changes made have improved the final result of the article which, I believe, is ready for publication. Congratulations for the work.

Author Response

We thank the reviewer for their detailed feedback on the manuscript and warm wishes. Addressing their concerns has substantially improved the validity of the study.

Round 3

Reviewer 1 Report

I am satisfied with the changes made by the authors. For me the manuscript is ready to be published in its current form.